# OpenReview forum: "Where Did It Go Wrong? Attributing Undesirable LLM Behaviors via Representation Gradient Tracing"
_ICLR.cc/2026/Conference — ICLR 2026 Poster_

### Official Review · Reviewer_Vu1g · 2025-10-31

**Soundness:** 1
**Presentation:** 3
**Contribution:** 2
**Rating:** 2
**Confidence:** 3

**Summary:**

The paper introduces Representation Gradient Tracing (RepT), a data attribution framework designed to identify which training examples caused specific undesirable behaviors in large language models, such as harmful outputs, backdoor activation, or factual contamination. Rather than operating in parameter space, RepT works in representation space, combining (1) the hidden state of the final prompt token and (2) the gradient of the first output token to construct a compact “signature” vector for each sample. Attribution is then performed via cosine similarity between signatures. The approach also supports token-level attribution by examining gradient-aligned influence scores across individual tokens.

**Strengths:**

The paper addresses a meaningful and timely challenge in model auditing: tracing the origins of harmful, biased, or otherwise undesirable behaviors in large language models. The shift from parameter-space attribution to representation-space attribution is conceptually coherent and aligns with recent evidence that internal activations capture semantically organized information. The proposed method is also computationally efficient, requiring only one backward pass per sample and storing compact representation signatures, which makes it more practical than influence-function-based approaches that require either Hessian approximations or historical gradient storage. Finally, the token-level attribution capability is a useful feature.

**Weaknesses:**

The evaluation setup substantially limits the strength and generality of the conclusions. All experiments are performed on synthetically constructed failure cases, where a very small and clearly defined set of poisoned or altered examples is inserted into an otherwise clean dataset. In these scenarios, the test prompts that elicit undesirable behavior are structurally and semantically very close to the corrupted training samples that caused them. This makes the attribution problem relatively straightforward, because the causal samples have strong, direct representational signatures that the method can recover via similarity-based comparison. In realistic settings, however, harmful, biased, or misleading behaviors rarely originate from a single or easily identifiable training example. Instead, they tend to arise from diffuse and distributed correlations spread across many heterogeneous data sources, often contaminated by noise, paraphrasing, or domain variation. The paper does not evaluate RepT under such circumstances, so the reported performance does not provide evidence that the method can handle real-world model failures.

The consistently near-perfect attribution accuracy reported across all models and all tasks is another point of concern. When a method achieves performance close to 1.0 in every condition, it suggests that the benchmarks may be systematically aligned with the method’s core retrieval mechanism. In this case, because the corrupted training examples closely resemble the test prompts in representation space, RepT may simply be identifying semantic similarity rather than performing genuine causal attribution. The paper does not examine how RepT behaves when the harmful or incorrect examples are paraphrased, stylistically altered, contextually embedded, or obfuscated, i.e., when surface-level correspondence no longer provides a direct signal. Without such tests, it is unclear whether the method is robust to more subtle or realistic forms of data contamination.

The method’s reliance on selecting a particular “phase transition” layer introduces another source of uncertainty. Although the paper proposes a heuristic based on representation similarity across layers, this procedure is not theoretically justified and is only lightly evaluated. There is no analysis of how stable the layer selection is across different domains, prompts, task types, or probing datasets, nor whether attribution accuracy deteriorates when the probing distribution diverges from the failure domain. Given that representation geometry varies significantly across models and training regimes, the absence of a stability study leaves open the question of whether the technique generalizes beyond the specific setups demonstrated.

Finally, the comparisons to baselines are not fully convincing due to insufficient detail about implementation constraints. Methods such as TracIn, LESS, and LoGra are highly sensitive to gradient storage policies, checkpoint frequency, normalization strategies, and available memory budgets. The paper reports performance differences but does not establish that these baselines were configured with matched computational constraints or tuned to competitive settings. As a result, it is difficult to determine whether RepT’s empirical advantage reflects an inherent methodological improvement or simply differences in experimental favorability.

**Questions:**

How does the method perform when the source of an undesirable behavior is distributed across many training instances, rather than localized to a single or a few poisoned samples?

Can the authors evaluate the method on naturally occurring hallucinations or biased generations, where the “ground truth causal sample” is unknown and must be approximated through retrieval or human labeling?

---

> ### Author Response · Authors · 2025-11-23
> **Response to Reviewer Vu1g (1/2)**
>
> Thank you for your detailed comments and suggestions. We believe there is a misunderstanding regarding the mechanism of RepT (specifically regarding semantic similarity) and have conducted comprehensive new experiments to clarify these points. Below are our responses to the issues raised:
>
> > **Q1: Critique on Synthetic Evaluation Setup (Is RepT just retrieving similar texts?)**
>
> RepT does more than just identify semantic similarity. To verify this, we compared RepT against BGE-M3, a state-of-the-art semantic embedding model, on Qwen2.5-7B-Instruct. If the task relied solely on semantic similarity, BGE-M3 should perform well.
>
> |Method|Harmful Tuning||Backdoor Attack||Ag $\rightarrow$ Na||Canada $\rightarrow$ Korea||
> |:-:|:-:|:-:|:-:|:-:|:-:|:-:|:-:|:-:
> ||Precision@100|AUPRC|Precision@100|AUPRC|Precision@100|AUPRC|Precision@100|AUPRC
> |RepT|**0.984**|**0.997**|**1.000**|**1.000**|**0.969**|**0.992**|**0.917**|**0.939**
> |BGE-M3|0.422|0.557|0.185|0.232|0.486|0.514|0.382|0.503
>
> The results demonstrate that the embedding model performs significantly worse than RepT, confirming that RepT is capturing mechanistic influence, not just semantic similarity. The "problematic" training data often differs semantically from the test data in our experiments.
>
> > **Q2: The paper does not examine how RepT behaves when the harmful or incorrect examples are paraphrased, stylistically altered, contextually embedded, or obfuscated. It is unclear whether the method is robust to more subtle or realistic forms of data contamination.**
>
> The training and testing data are already paraphrased or mutated to reduce potential semantic bias. For example, in the harmful data identification task, we use data from LAT [1], which introduces extensive paraphrased mutations based on harmful sources such as AdvBench and HarmBench. In the knowledge contamination attribution task, we prompt GPT-4o mini to generate more diverse examples covering different aspects of the altered knowledge. As shown in Q1, embedding models that rely purely on semantic similarity perform significantly worse than RepT, confirming that the "problematic" training data often differ in phrasing or context from the test prompts.
>
> To strengthen the validation that RepT is robust to more subtle or realistic forms of data contamination, we use GPT-4o mini to generate highly diverse/obfuscated paraphrases (1,000) of our original test data for each task.
>
> |Method|Harmful Tuning||Backdoor Attack||Ag $\rightarrow$ Na||Canada $\rightarrow$ Korea||
> |:-:|:-:|:-:|:-:|:-:|:-:|:-:|:-:|:-:
> ||Precision@100|AUPRC|Precision@100|AUPRC|Precision@100|AUPRC|Precision@100|AUPRC
> |RepT|**0.986**|**0.998**|**0.998**|**1.000**|**0.922**|**0.989**|**0.912**|**0.941**
> |BGE-M3|0.122|0.231|0.145|0.202|0.246|0.314|0.212|0.335
>
> RepT remains robust even under heavy paraphrasing, while semantic matching method fails.
>
> [1] Sheshadri, Abhay, Aidan Ewart, Phillip Guo, Aengus Lynch, Cindy Wu, Vivek Hebbar, Henry Sleight et al. "Latent adversarial training improves robustness to persistent harmful behaviors in llms." arXiv preprint arXiv:2407.15549 (2024).
>
> > **Q3: Layer Selection (phase transition point) Stability**
>
> As shown in our ablation study (Figure 3, middle panel), RepT is highly stable across layers. While we provide a heuristic for the "optimal" layer (phase transition), the performance curve is broad and flat. Even selecting a sub-optimal layer (e.g., the last layer) yields competitive results. The method is not brittle to this hyperparameter.

---

> ### Author Response · Authors · 2025-11-23
> **Response to Reviewer Vu1g (2/2)**
>
> > **Q4: Fairness of Baseline Comparisons**
>
> We strictly followed the official implementations and hyperparameters for all baselines (details in Appendix), and adopted the same training protocol for fair comparison. For TracIn, we even improved the baseline by adding Layer Normalization (TracInLN), which significantly boosted its performance compared to the vanilla version as suggested in RapidIn. For RapidIn and LESS, we apply the same settings including the number of random shuffle and the random projection dimension as recommended in the original paper. For LoGra, we used the official framework which logs gradient information during training.
>
> > **Q5: How does the method perform when the source of an undesirable behavior is distributed across many training instances, rather than localized to a single or a few poisoned samples?**
>
> We have to emphasize that RepT is designed to handle group effects naturally. The output of RepT is **not a single sample**, but a **ranked list** based on influence scores. If a behavior is caused by a group of interacting samples, RepT will assign high influence scores to all of them, placing them at the top of the ranking. For example, in our harmful data identification experiment, we successfully identified multiple harmful samples that collectively degrade safety.
>
> > **Q6: Can the authors evaluate the method on naturally occurring hallucinations or biased generations, where the “ground truth causal sample” is unknown and must be approximated through retrieval or human labeling?**
>
> We thank the reviewer for raising this question. We design two new scenarios: (1) fine-tuning with benign data while compromising model's safety alignment; and (2) detecting semantic backdoor poisons in the training set that may induce biased generation. Since explicit ground truth is unavailable, we evaluate instead the change in the re-trained model's performance after removing the top influential/problematic samples identified by RepT from the training set.
>
> **Benign Fine-tuning**: We conducted an experiment following [2, 3]. We fine-tuned a model using only benign data to compromise the model's safety alignment. We then identified influential samples using attribution methods and removed them to see if the ASR dropped.
>
> |Method|ASR (original)|ASR (removing identified influential data)|
> |:-:|:-:|:-:
> |TracIn|0.894|0.811
> |TracInLN|0.894|0.743
> |RapidIn|0.894|0.815
> |LESS|0.894|0.421
> |LoGra|0.894|0.623
> |RepT|0.894|0.149
>
> The results show that attribution methods successfully identify "problematic" data where removing them significantly reduces the ASR.
>
> **Semantic Backdoor**: We follow the work [4] to construct semantic backdoor training data, inducing the model to produce biased recommendations when prompted with food-related queries. We then use attribution methods to identify influential samples and remove them to evaluate whether the TSR (trigger success rate) decreases.
>
> |Method|TSR (original)|TSR (removing identified influential data)|
> |:-:|:-:|:-:
> |TracIn|0.974|0.968
> |TracInLN|0.974|0.935
> |RapidIn|0.974|0.971
> |LESS|0.974|0.808
> |LoGra|0.974|0.769
> |RepT|0.974|0.332
>
> The results show that attribution methods successfully identify "problematic" data where removing them significantly reduces the TSR.
>
> [2] He, L., Xia, M., & Henderson, P. (2024). What is in your safe data? identifying benign data that breaks safety. arXiv preprint arXiv:2404.01099.
>
> [3] Zhao, W., Li, Z., Li, Y., & Sun, J. (2024). Unleashing the Unseen: Harnessing Benign Datasets for Jailbreaking Large Language Models. arXiv preprint arXiv:2410.00451.
>
> [4] Min, N. M., Pham, L. H., Li, Y., & Sun, J. (2025). Propaganda via AI? A Study on Semantic Backdoors in Large Language Models. arXiv preprint arXiv:2504.12344.

---

### Official Review · Reviewer_5E2k · 2025-11-01

**Soundness:** 3
**Presentation:** 3
**Contribution:** 3
**Rating:** 6
**Confidence:** 3

**Summary:**

The paper introduces a novel and efficient framework that diagnoses a range of undesirable LLM behaviors by analyzing representation and its gradients, which operates directly in the model’s activation space to provide a semantically meaningful signal linking outputs to their training data. Overall, I think the method is reasonable and the experimental results are promising.

**Strengths:**

- Reasonable method. The idea of using gradients and activations for tracing relevant training data sounds promising.
- Good results. The experimental results are good, demonstrating the advantages of the proposed method.
- Comprehensive ablation study. Ablation study shows the effectiveness of each component.

**Weaknesses:**

- Baseline selection. I think in some settings, the displayed baselines may be not appropriate. For example, for harmful data identification, a straight forward approach is to use safety classifiers. Additionally, for backdoor data detection, there are various baselines specific for backdoor detection. I think these baselines should be stronger than the current baselines.
- Scalability. The paper includes an interesting experiment part for knowledge contamination detection. While the experiment shows that the method works well for a small fine-tuning dataset, I think knowledge contamination is more important when applied on pretraining data. However, the current method seems hard to work on pretraining data, as it requires model inference on each training sample, whose cost is too high.

**Questions:**

Typo: line 237 effective->effectivenes

---

> ### Author Response · Authors · 2025-11-23
> **Response to Reviewer 5E2k**
>
> Thank you for your positive and insightful comments. We address the concerns as below.
>
> > **Q1: Baseline Selection**
>
> We respectfully clarify that detection and attribution serve fundamentally different goals. Detection classifiers (e.g., Llama Guard) aim to predict labels (e.g., harmful or harmless), while data attribution focuses on identifying the causal origin (i.e., which training sample caused this behavior?). A training sample can be mechanistically responsible for a harmful output while being semantically benign itself. In such cases, safety classifiers fail completely because the data itself is not harmful.
>
> To demonstrate this, we conducted an experiment following [1, 2]. We fine-tuned a model using only benign data to compromise the model's safety alignment. We then identified influential samples using attribution methods and a classifier Llama-Guard-3-8B, and removed them to see if the ASR dropped.
>
> |Method|ASR (original)|ASR (removing identified influential data)|
> |:-:|:-:|:-:
> |RepT|0.894|0.149 (Significant Safety Improvement)
> |LESS|0.894|0.421
> |Llama-Guard-3-8B|0.894|0.892 (Fails to identify causal data)
>
> As the training data contains no explicit harmful content, Llama-Guard-3-8B detects merely nothing, while attribution methods successfully identify "problematic" data where removing them significantly reduces the ASR.
>
> Similarly, most backdoor defenses focus on trigger inversion or model sanitization. They do not perform sample-level data attribution to identify the exact poisoned samples in the training set, especially when triggers are semantically meaningful and diverse.
>
> We agree that external classifiers may be effective for certain specific tasks, but we emphasize that our data attribution approach is designed as a general framework that applies across diverse scenarios, rather than being limited to any single one.
>
> [1] He, L., Xia, M., & Henderson, P. (2024). What is in your safe data? identifying benign data that breaks safety. arXiv preprint arXiv:2404.01099.
>
> [2] Zhao, W., Li, Z., Li, Y., & Sun, J. (2024). Unleashing the Unseen: Harnessing Benign Datasets for Jailbreaking Large Language Models. arXiv preprint arXiv:2410.00451.
>
> > **Q2: Scalability to Pre-training Data**
>
> We agree that applying attribution to pre-training data (trillions of tokens) is a significant challenge. However, RepT offers a feasible solution compared to other gradient-based attribution methods in this scenario.
>
> RepT requires running inference (forward/backward) on the training data only once to cache the signatures. The computational cost is equivalent to one epoch of training. For pre-training datasets, this is computationally expensive but feasible for the entity training the model. In contrast, influence functions require prohibitive Hessian computations which is impossible at this scale. Other gradient-based methods require computing and compressing the full parameter-gradient vector, which is also impractical at this scale.
>
> We provide an experiment on the full Alpaca-cleaned dataset (around 7.8 million tokens) with a 5% poisoning ratio in backdoor poisoning detection task to evaluate scalability on Qwen2.5-7B-Instruct. We also introduce a 16-bit storage variant to further reduce overhead.
>
> | Dataset Size | 5k Samples (0.75M Tokens) | 50k Samples (7.8M Tokens) | 50k Samples (7.8M Tokens), 16-bit
> | :--- | :---: | :---: | :---: |
> | Precision@100 | 1.000 | 0.992 | 0.986
> | AUPRC | 1.000 | 0.998 | 0.991
> | Caching Time | 0.36h | 3.91h | 2.02h
>
> The results demonstrate that RepT maintains high precision even as the training data size grows without the significant degradation, and the caching time remains linear and manageable.

---

### Official Review · Reviewer_TAkS · 2025-11-01

**Soundness:** 2
**Presentation:** 2
**Contribution:** 2
**Rating:** 4
**Confidence:** 3

**Summary:**

This paper introduces Representation Gradient Tracing (RepT), a novel framework that diagnoses undesirable LLM behaviors by analyzing representations and their gradients in the activation space rather than the parameter space. RepT constructs compact signature vectors by concatenating the last prompt token's hidden state with the first response token's gradient, enabling efficient sample-level and token-level attribution through cosine similarity. Experiments across harmful content identification, backdoor detection, and knowledge contamination show RepT achieves near-perfect precision (≈100% auPRC) while requiring 1000× less memory and significantly faster computation than gradient-based baselines.

**Strengths:**

1. I like the task and it is well motivated. I think attributing LLM behavior is quite important and this paper provides a practical and efficient method.
2. The shift from parameter space to representation space is sound and addresses the limitations of gradient-based methods (high dimensionality, noise, semantic gap). The design of signature vectors—combining H_last (contextual understanding) and g_H_first (predictive direction)—elegantly captures the causal link between inputs and outputs.

**Weaknesses:**

1. **Missing Important Intuitive Baselines**: The paper lacks comparison with simpler, more intuitive baseline methods. For instance, one could directly use a lightweight embedding model (e.g., the popular bge-m3) to compute semantic similarity between training samples and problematic test cases, which does not rely on gradient-based approaches at all. Since Appendix A provides insufficient implementation details for existing baselines, it remains unclear for me whether such embedding-based methods were considered or why they were excluded from evaluation.
2. **Insufficient Scalability Analysis**: The paper does not adequately address how RepT scales along two critical dimensions of LLM growth. First, as model size increases, the hidden layer cache requires proportionally more storage and computation—how does this affect feasibility for 70B+ parameter models? Second, and more critically, how does performance degrade with massive training corpora? The method's linear time/space complexity with respect to training set size (O(N)) makes it seemingly impractical for pretraining scenarios with 10T+ tokens. Moreover, attribution accuracy likely decreases as the training set grows larger and more diverse—does the method still precisely identify causal samples when N reaches millions or billions? These scaling properties are not empirically studied.
3. **Lack of Analysis for Complex Attribution Scenarios**: The evaluation assumes each bad case can be traced to a single training sample, but real-world failures are often more nuanced. First, many undesirable behaviors emerge from the **interaction of multiple training samples**—none problematic individually but harmful in combination. Can RepT detect such collective effects? Second, some failures stem from the **absence of appropriate training data** rather than the presence of bad data. In such cases, would the method force-match to irrelevant samples, producing misleading attributions? The paper does not discuss these failure modes or provide diagnostic guidance for practitioners.

**Questions:**

See above

---

> ### Author Response · Authors · 2025-11-23
> **Response to Reviewer TAkS (1/2)**
>
> Thank you for your valuable comments and constructive suggestions. We appreciate the recognition of our method's motivation and efficiency. Below are our responses to the issues raised:
>
> > **Q1: Missing Important Intuitive Baselines (Comparison with Embedding Models)**
>
> We appreciate the suggestion about this intuitive baseline. While our primary focus was on mechanistic influence where a training sample can be mechanistically responsible for an output while being semantically unrelated to it, we agree that comparing with semantic embedding models strengthens the validation. We have added experiments comparing RepT with BGE-M3 across all tasks on Qwen2.5-7B-Instruct.
>
> |Method|Harmful Tuning||Backdoor Attack||Ag $\rightarrow$ Na||Canada $\rightarrow$ Korea||
> |:-:|:-:|:-:|:-:|:-:|:-:|:-:|:-:|:-:
> ||Precision@100|AUPRC|Precision@100|AUPRC|Precision@100|AUPRC|Precision@100|AUPRC
> |RepT|**0.984**|**0.997**|**1.000**|**1.000**|**0.969**|**0.992**|**0.917**|**0.939**
> |BGE-M3|0.422|0.557|0.185|0.232|0.486|0.514|0.382|0.503
>
> RepT significantly outperforms the embedding baseline. This confirms that RepT captures causal links even when the training data is not semantically identical to the test query, while embedding models fails.
>
> > **Q2: Scalability Analysis (Model Size and Data Size)**
>
> We thank the reviewer for raising these questions regarding scalability. We address both dimensions below:
>
> **1. Scalability with Model Size (7B vs. 70B)**
>
> As detailed in Section 5 **Table 4**, RepT already demonstrates efficient scaling to Llama2-70B where baselines fail..
> * **Storage:** Unlike gradient-based methods (e.g., Influence Functions, TracIn) where storage of the gradient vector grows linearly with the parameter count, RepT's storage depends only on the hidden state dimension $d$. When scaling from Llama2-7B to Llama2-70B, $d$ only increases from 4096 to 8192 (a $2\times$ increase), whereas parameters increase by $10\times$. This makes RepT's memory increase negligible.
> * **Computation:** RepT requires only a single forward and backward pass per sample. In contrast, baselines often require expensive operations like computing the Inverse Hessian or performing random projections on massive parameter vectors to fit in memory. As shown in Table 4, RepT processes Llama2-7B (full fine-tuning) in just **0.4 hours** on 0.75M tokens, whereas baseline methods either run out of memory or require hundreds of hours to manipulate gradient vectors. For Llama2-70B, gradient-based methods are impractical for fully fine-tuned models due to OOM or massive compute costs, RepT can efficiently handle the attribution in around 4 hours on 0.75M tokens.
>
> **2. Scalability with Training Data Size**
>
> To empirically evaluate RepT's stability as data size increases, we extended our experiments to the full Alpaca-cleaned dataset (around 7.8 millions of tokens) as the clean source with 5% poisoning ratio to evaluate RepT's performance on the backdoor poisoning detection task for Qwen2.5-7B-Instruct.
>
> | Dataset Size | 1k Samples (0.15M Tokens) | 5k Samples (0.75M Tokens) | 10k Samples (1.5M Tokens) | 50k Samples (7.8M Tokens) | 50k Samples (7.8M Tokens) bfloat16 |
> | :--- | :---: | :---: | :---: | :---: | :---: |
> | Precision@100 | 1.000 | 1.000 | 0.997 | 0.988 | 0.986
> | AUPRC | 1.000 | 1.000 | 0.999 | 0.992 | 0.992 | 0.991
> | Caching Time | 0.07h | 0.36h | 0.74h | 3.91h | 2.02h
>
> The results demonstrate that RepT maintains high precision even as the training data size grows without the significant degradation. RepT demonstrates strong scalability even at the million-token level.

---

> ### Author Response · Authors · 2025-11-23
> **Response to Reviewer TAkS (2/2)**
>
> > **Q3: Lack of Analysis for Complex Attribution Scenarios**
>
> We thank the reviewer for insightful points in the complex attribution scenarios.
>
> * **Interaction of Multiple Samples (Group Effects)**: RepT is designed to handle group effects naturally. The output of RepT is **not a single sample**, but a **ranked list** based on influence scores. If a failure arises from a group of interacting samples, RepT will assign high influence scores to all of them, placing them at the top of the ranking. For example, in our harmful data identification experiment, we successfully identified multiple harmful samples that collectively degrade safety.
>
> * **Absence of Data**: We agree that attribution methods cannot "find" data that does not exist. However, RepT provides a diagnostic signal for this scenario through the magnitude of the influence score. If bad data are present, RepT will assign them high influence scores. If a model fails due to missing specific knowledge, RepT will produce low maximum influence scores, indicating that no specific training sample is mechanistically responsible. This allows users to distinguish between "misaligned data" (high influence score) and "missing data" (low influence score).
>
> We added experiments on safety alignment: (1) harmful tuning of Qwen2.5-7B-Instruct (aligned model) to compromise its safety alignment, and (2) fine-tuning Qwen2.5-7B (base model) with clean data but without any safety-related data. We then computed the mean influence score of the top 100 candidates identified by RepT.
>
> | Model | ASR | Mean Influence Score | Conclusion
> | :--- | :---: | :---: | :---: |
> | Qwen2.5-7B-Instruct (harmful tuning) | 0.992 | 0.837 | High score indicates bad data caused the behavior.
> | Qwen2.5-7B (absence of safe data) | 1.000 | 0.412 | Low score indicates no specific data is responsible.
>
> While both models exhibit unsafe behaviors (high ASR), RepT distinguishes the cause. When harmful behaviors are induced by harmful training data, RepT assigns relatively high influence scores to the identified samples. In contrast, in the absence of safe data scenario, RepT assigns only small influence scores to the identified examples. We will add a discussion in the revision on using score thresholding as a diagnostic tool for the failures stem from the absence of appropriate training data.

---

### Meta-Review · Area_Chair_T5q3 · 2026-01-05

**Summary:**

This paper proposes a representation gradient tracing (RepT) for data attribution at different levels (data sample and token) and shows applications in several safety-related settings. The original review comments were concentrated on the practicality and complexity of the evaluation datasets, and I believe the authors' rebuttal has successfully addressed the major concerns.

While the original ratings are mixed, I would advocate acceptance.

**Reviewer Concerns:**

I believe all major concerns were addressed.

**Reviewer Scores:**

I believe all reviewers would bump up their scores.

---

### Decision · Program_Chairs · 2026-01-26

Accept (Poster)